# Joint Dysfunctionality Alleviation along with Systemic Inflammation Reduction Following Arthrocen Treatment in Patients with Knee Osteoarthritis: A Randomized Double-Blind Placebo-Controlled Clinical Trial

**DOI:** 10.3390/medicina58020228

**Published:** 2022-02-02

**Authors:** Ramin Goudarzi, Peter Thomas, Sandra Ryan

**Affiliations:** 1Division of Research and Development, Pharmin USA, LLC, San Jose, CA 95128, USA; ramin.goudarzi@yahoo.com; 2School of Medicine, University of California, San Francisco, CA 94143, USA; peterthomasuscf@gmail.com; 3Independent Scholar, 24298 El Pilar, Laguna Niguel, CA 92677, USA

**Keywords:** avocado, LISOK, osteoarthritis, SF-20, soybean, unsaponifiables

## Abstract

*Background and objectives*: Many mediators and cytokines are involved in the pathogenesis of osteoarthritis (OA). Some of these cytokines are spontaneously expressed by cultured fibroblast-like synoviocytes. Therefore, using serum samples, the efficacy and the effects of avocado/soy unsaponifiables, ASU, (Arthrocen) on cytokine changes were assessed in patients with knee OA (KOA). *Materials and Methods:* Experimental procedure: A randomized, double-blind, placebo-controlled clinical trial was conducted on patients with a diagnosis of mild to moderate OA who received either Arthrocen 300 mg/day (*n* = 61) or placebo (*n* = 58) for 3 months. Data collection was performed using questionnaires including the Western Ontario and McMaster Universities osteoarthritis index (WOMAC), 20-item short form survey (SF-20), Lequesne index of severity for osteoarthritis of the knee (LISOK), and three visual analog scales (VASs) as pain quality indices. The serum levels of interleukins 2 (IL-2), IL-4, IL-10, IL-17α, and TNF-α were measured using an ELISA reader. *Results:* Both quality of life indices, pain sensation and scored by specialists (as VASs), respectively, including WOMAC and SF-20, as well as joint dysfunctionality symptoms assessed by physicians were significantly improved (*p* < 0.05) in OA patients receiving Arthrocen. The serum levels of anti-inflammatory interleukins 4 and 10 were also augmented, while levels of inflammatory IL-17 and TNF-ɑ cytokines were decreased significantly (*p* < 0.05) compared with the control groups during the 3- and 6-month treatment. *Conclusions:* Arthrocen consumption may increase the quality of life in OA patients through amelioration of inflammation and improvement of functional activities without any adverse effects in the long term.

## 1. Introduction

Human osteoarthritis is a degenerative disorder involving activation of the immune system accompanied by the production of pro-inflammatory cytokines that play a significant role in the destruction of articular cartilage and disease progression [1]. Although OA is the most common musculoskeletal disorder causing major social and health problems throughout the world, its treatment is challenging and its pathophysiology is still unknown [2]. The onset of OA results from a combination of risk factors, including aging, overuse, limb malformation, genetic disorders, and metabolic problems (obesity, immune responses, diabetes mellitus) [3]. The hip and knee are the major large joints affected by OA [4].

The current treatment options for OA vary from non-pharmacological and pharmacological strategies to joint replacement [5]. The partial efficacy, besides adverse effects and potential toxicities of current drugs such as non-steroidal anti-inflammatory drugs (NSAIDs) and opioids, limit their clinical application [6]. Although knee replacement provides a substantial improvement in symptoms in the majority of cases, its associated costs and risks are other concerns [7,8]. Altogether, these factors have led to a serious and growing need for new medical treatments, especially for pain and inflammation. 

On the other hand, there is an increasing tendency toward using herbal extracts as a way to treat OA due to their time-tested safety and efficacy [9]. In particular, the avocado/soy unsaponifiable (Arthrocen) supplement, a phytosterol extract with a commercially available form known as Arthrocen, is classified as prescription human medicine or supplement according to the regulatory guidelines set by each country’s health ministry [10]. Phytosterols such as dihydrocholesterol, campesterol, stigmastanol, and b-sitosterol are the major components of nutraceuticals composed of ASU which are obtained from avocado and soy unsaponifiables in a ratio of 1:2, respectively [11]. Oral ASU is administrated commonly at a dose of 300 mg/day for the therapeutic periods from 3 months to 3 years to improve the functional ability and joint pain. At the clinical level, ASU reduces pain and stiffness and improves joint function, resulting in decreased dependence on analgesics [12]. So far, FDA-approved ASU has not generally shown any significant adverse events at doses of 300–600 mg/day in experimental or meta-analysis reports. Arthrocen efficacy and safety during and after treatment have been assessed in various randomized, double-blind, multicenter trials in patients with symptomatic knee or hip OA [12]. 

In terms of cellular mechanisms of OA, it is mainly started by sequential inflammation and then local inflammation in the damaged joints that is associated with systemic inflammation markers [13]. It is proposed that systemic inflammation has a key role in osteoarthritis progression that is associated with pain severity, muscle weakness, and poor proprioception in patients’ OA [14,15]. While inflammation plays a major role in the pathology of OA, the effects of ASU on molecules implicated in inflammation are still poorly explained in patients with OA. There are several reports of the stimulatory effects of ASU on anti-inflammatory and anabolic mediators as well as its inhibitory effects on inflammatory and catabolic mediators [10,16,17,18]. Previous reports have shown that ASU can reduce the production of pro-inflammatory mediators such as IL-1, IL-6, IL-8, and TNFα in various cell types [16,19,20]. Moreover, many of these cytokines are involved in the pathogenesis of OA [21]. These findings indicate that while sample preparation from the cartilage or knee is invasive and difficult, evaluation of the serum levels of these cytokines may be more valuable, especially for clinical studies and diagnostic implications. Therefore, the present study was conducted to examine the effect of ASU on the health condition and quality of life of patients with knee OA as well as their serum levels of interleukin 4 and 10 (as the main anti-inflammatory cytokines), TNF-α (as the main inflammatory cytokine), and IL-17 (as a signature cytokine of the Th17 subset).

## 2. Materials and Methods

### 2.1. Study Design and Patients 

This study was conducted in Tehran hospitals according to the ethical guidelines of the Iranian Ministry of Health (ethical code and date: IR.MOH 1396.575, 21 March 2018). In our research, we used “Consolidated Standards of Reporting Trials (CONSORT) 2010” that was adapted from http://www.consort-statement.org/consort-2010 (accessed on 29 January 2018). Initially, the study protocol was evaluated and approved by the Ethics Committee of the Iranian Ministry of Health. The tenets of the Declaration of Helsinki and Nuremberg Protocol were observed in all stages of the study.

A double-blind randomized clinical trial was performed on patients that suffered from OA in hospital clinics of Tehran between March 2018 and December 2019. All patients were individually informed about the study protocol and randomized placebo-Arthrocen prescription, and consent was obtained from each patient before inclusion in the study. The inclusion criteria were as follows: (a) mild to moderate degenerative knee OA, (b) bilateral OA, and (c) age under 65 years. The OA of the knee was diagnosed according to the clinical and radiological findings and self-reports of pain in patients with a mild to moderate active movement index (minimum of 40 mm on a 100-mm of VAS) based on the American College of Rheumatology (ACR) criteria [22]. Patients with secondary OA from trauma; rheumatoid arthritis; drug sensitivity; ESR > 20; heart, kidney, or hepatic failure; malabsorption disorders; history of psychological disease; any intra-articular injection of corticosteroid during the last 3 months; and a BMI more than 35 or less than 20 were excluded from the study.

Eligible patients presenting to the clinics were randomly assigned to treatment and control groups. Patients in the treatment group received Arthrocen (Pharmin USA, LLC, SanJose, CA, USA) capsules containing 300 mg ASU for 6 months, while placebo was used in the control group d. The duration of treatment was three months (once daily) in both groups. Before the clinical trial, a completely randomized sample of patients was used as a basis for the comparison of post-test data.

The patients were assessed before and after the experiment in three ways: (a) measuring the serum levels of systemic cytokines, (b) visiting patients by an experienced team of rheumatologists, and (c) using several questionnaires which are described in the below section. A brief diagram of the study procedure related to clinical and preclinical assessments is given in Figure 1.

### 2.2. Inflammatory Cytokines Evaluation

Blood samples were collected from each participant following a standardized protocol and according to clinical practice. Briefly, serums were separated from the whole blood by 2000 rpm centrifugation. Then, the levels of IL-4, IL-10, IL-17α, and TNF-α cytokines were measured in the serum samples using the sandwich Elisa method according to the manufacturer’s instructions (eBioscience, San Diego, CA, USA).

### 2.3. Clinical Assessments 

In brief, in this study, the patient was visited by an experienced team of rheumatologists that assessed factors such as internal and lateral joint tenderness, swelling, patella crepitus, crepitus, and flexion contracture. Moreover, questionnaires included the Western Ontario and McMaster Universities osteoarthritis index (WOMAC) [23], 20-item short form survey (SF-20) [24], Lequesne index of severity for osteoarthritis of the knee (LISOK) [25], a pain severity index on a 100-mm visual analog scale (VAS) reported by patients, and two VAS indices for overall health condition reported by both the rheumatologists and the patients. WOMAC is widely used to evaluate pain, stiffness, physical function, and psychosomatic relations to OA. The SF-20 survey is used to assess the quality of life of patients by examining six aspects of their lives, including physical functioning, role functioning, social functioning, mental health, current health perceptions, and pain. LISOK looks at pain, maximum distance walked, and activities of daily living. The valuation of the items in these questionnaires is such that the more SF-20 value, the lower the WOMAC value, the lower the LISOK value, the lower the VAS value (for pain severity), and the higher the VAS value (for overall health condition) reflecting a more favorable health condition. 

## 3. Statistical Analysis 

All statistical analyses were performed in open-source statistical software R (v 3.4.0), a language and environment for statistical computing [26]. In the data preparation process, outliers were detected and removed carefully using Tukey’s method, which identifies the outliers ranging above and below the 1.5 interquartile range (IQR) [27]. In order to compare pre-test and post-test data, paired and unpaired t-tests were applied to questionnaires and serological data, respectively. If the data did not have a normal distribution, Wilcoxon signed-rank test and Mann–Whitney test (Wilcoxon rank-sum test) was used for paired and unpaired samples, respectively. In addition, the chi-square test was used to test the association between nominal variables including some demographic data. *p*-value thresholds of less than 0.05 and 0.10 were considered as significant and suggestive levels, respectively.

## 4. Results 

### 4.1. Systemic Anti-Inflammatory Effect of ASU

To evaluate the effect of Arthrocen on inflammatory cytokines, we measured four mediators in the serum samples of the participants. Data analysis revealed that the serum levels of inflammatory factors including IL-17 and TNF-α decreased after Arthrocen treatment compared to the control and receiving placebo groups, while these factors were similar before and 3 and 6 months after placebo administration (*p* < 0.05) (Figure 2A,B).

By contrast, the results showed a significant increase in the serum levels of anti-inflammatory IL-10 and IL-4 compared to the relative control groups; however, the placebo group was not significantly different from the control group. In brief, the results showed the effect of Arthrocen on anti-inflammatory factors (Figure 2C,D).

### 4.2. OA Improvement in Patients Receiving Arthrocen

A total number of 140 OA patients were recruited based on the inclusion criteria, but only 116 of them continued the study to the end, including patients in the Arthrocen and placebo groups. There was no significant difference in demographic factors, VAS, WOMAC, and LISOK scores between the study groups at baseline. We examined the effects of Arthrocen and placebo on OA patients’ health condition through a comprehensive examination conducted by a team of board-certified rheumatologists and pharmacologists. The outcome of this assessment showed significant clinical benefits in both treatment groups compared to pre-test (*p* < 0.05). However, there was no significant difference between placebo and ASU groups before treatment (Table 1). 

Analysis of the results of examinations before and after treatments showed a significant reduction in internal or lateral tenderness in the Arthrocen (*p* < 0.001) and placebo (*p* < 0.05) groups (Table 1). The patella crepitus and joint shrug signs improved significantly (*p* < 0.001, respectively) after Arthrocen treatment. However, the difference between the two groups was not significant (*p* > 0.05). 

According to Table 2, the SF20, WOMAC, LISOK, and VAS showed no significant differences between the two groups of patients at the baseline (before treatments). The SF20 score, as an indicator of the quality of life, increased significantly in the Arthrocen group (*p* < 0.001). The results of the WOMAC and VAS for pain severity showed similar results; however, despite a significant decrease after Arthrocen treatment in both groups (*p* < 0.001 and *p* < 0.001, respectively), no difference was found between the placebo and Arthrocen groups. The LISOK (as an indicator of the health status and living activities) did not change significantly after treatments in both groups. Although two VAS indicators for the overall health status of patients reported by patients and physicians showed significant improvements (*p* = 0.001 and *p* < 0.01, respectively) in patients treated with Arthrocen, these indicators showed no significant difference with the placebo groups (*p* = 0.06, *p* = 0.22, respectively). Therefore, the health status improved significantly after the intervention, and Arthrocen had a greater effect on the improvement of the patients compared to placebo. 

## 5. Discussion 

The present study was conducted to examine the effect of Arthrocen on the patients’ quality of life, activities of daily living, and pain and their relationship with inflammatory cytokines in patients with knee OA, as these molecules are thought to play a leading role in the pathogenesis of OA. It is worth mentioning that the role of inflammatory and anti-inflammatory cytokines in the pathogenesis of OA is not clear, which warrants further research [28]. Our findings described that the severity of OA assessed in terms of intensity of pain and the joint problem can be strongly associated with the increased level of inflammatory cytokines in the group of patients. Meanwhile, Arthrocen induces anti-inflammatory cytokine factors that may be associated with improvement of SF-20, WOMAC, and VAS (pain) indices in the patients with advanced OA. During the trial, no side effects were also reported.

Based on the results, Arthrocen significantly improved the quality of life, activities of daily living, and pain from OA; moreover, two important anti-inflammatory cytokines including IL-4 and 10 increased markedly in the serum samples of Arthrocen patients. Although IL-10 increased in the placebo-treated group, this may be more related to the use of NSAIDs, especially diclofenac or acetaminophen codeine during intolerable pain, which was more common in the placebo-treated patients [29].

Cytokines are classified according to their inflammatory or anti-inflammatory role [30]. Interestingly, both IL-4 and IL-10, whose serum levels increased significantly in OA patients after Arthrocen ingestion, are anti-inflammatory cytokines [31]. Interleukin-4 and 10 mainly exert their effects through the suppression of other inflammatory factors [31]. Since these two cytokines inhibit the joint production of destructive enzymes (e.g., metalloproteinase), they can prevent the destruction of cartilage and its surrounding tissue [32,33,34,35,36]. In practice, it has also been proven that IL4-10 boosts direct and indirect structural cartilage repair in osteoarthritis [37]. 

Human chondrocytes synthesize IL-10 and express IL-10R on their surface. Since IL-10 inhibits IL-1 and TNF-alpha expression, its upregulation in osteoarthritic chondrocytes may counteract the detrimental effects of these catabolic cytokines. However, the functions of IL-10 in cartilage may go beyond those activities established in the immunological setting. The high levels of IL-10 in fetal cartilage, as an actively growing tissue, suggest that IL-10 may play a role in controlling chondrocyte metabolism under physiological conditions, which is very important for normal function [38].

Previous studies on ASU showed that ASU suppresses TNF-α, IL-1β, COX-2, NOS gene expression, and PGE2 and nitric oxide production in human articular chondrocytes and monocyte/macrophages [39]. In the present study, ASU increased the serum level of IL-10. It has been shown that IL-10 stimulates the synthesis of IL-1β and TNF-α antagonists and significantly reduces the production of these two cytokines [34]. In articular cells, IL-1β stimulates the synthesis of other cytokines such as TNF-α, IL-6, IL-8, and CCL5 [39]. Therefore, the synthesis of these molecules may decrease when Arthrocen suppresses the IL-1β production pathway. The previously reported blockade effects of ASU on metalloproteinases and collagenases such as TIMP-1, MMP-2, and MMP-3 could be due to an increase in levels of interleukin 4 and 10, which was also observed in the present study [40,41].

In this study, placebo had no marked effect on the health status of patients except for the VAS reported by the patient. In contrast, a meta-analysis of 35 randomized placebo-controlled trials by Machado et al. found that, in comparison with placebo, NSAIDs did not provide a clinically important effect on spinal pain [42]. 

It was tried to collect the maximum possible data through questionnaires, which could reduce the reliability and precision of the data from the questionnaires because a large number of questionnaires and questions may be tedious for patients. Therefore, a clinical evaluation was carried out by experts to ensure the quality of the questionnaire data. Fortunately, except for the SF-20 index, the results of other questionnaires were almost similar to clinical evaluations by experts. The reason for this mismatch can be the complexity of SF-20 questions compared to other scales. In this trial, the results of the questionnaires and clinical evaluations showed a significant difference between the ASU and placebo groups in terms of the health status of patients as well as pain reduction, which were consistent with the anti-inflammatory effect of ASU administration on reducing TNF-α and IL-17 and increasing IL-10, and IL-4 blood levels in comparison with placebo in both early and chronic stages of drug administration.

The limitations of our study would be taken into account when interpreting our results. At first, the interference events such as smoking and pollution may involve in the assessment of systemic inflammation or pain within a patient. Secondly, although widely used to assess the intensity of pain in clinical studies, the measurement of pain using a VAS is influenced by a variety of factors associated with the perception of pain [14,15]. Nevertheless, pain is always subjective, and the observed associations provide evidence for this subjective phenomenon to be better associated with the markers of systemic inflammation than any of the objective markers of OA assessed in our study. Thirdly, the lack of local inflammatory characteristics is a limitation in which neither knee joint nor synovitis biomarkers were assessed. Finally, the pathology behind inflammation in additional joint regions remains indefinable.

## 6. Conclusions

These findings reasonably advocate the anti-inflammatory properties of Arthrocen that may affect the joint pain and dysfunctionality improvement as well as the quality of life in OA patients. Improving joint function and pain relief can cause patient activity, which is important for public health concerning the recovery of quality in lifestyle. At the clinical level, Arthrocen may prove to be an effective therapeutic agent with minimum side effects that prevents progression of OA symptoms.

## Figures and Tables

**Figure 1 medicina-58-00228-f001:**
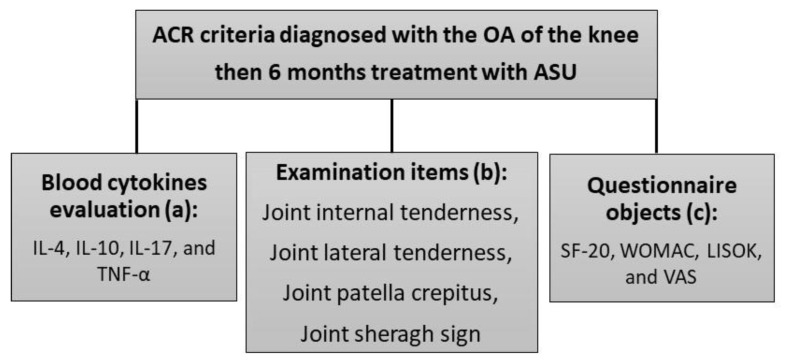
The diagram of the study procedure related to the diagnosis of the patient’s OA and clinical and preclinical assessments. (a) Preclinical tests. (b) Clinical evaluations. (c) Patient’s reports.

**Figure 2 medicina-58-00228-f002:**
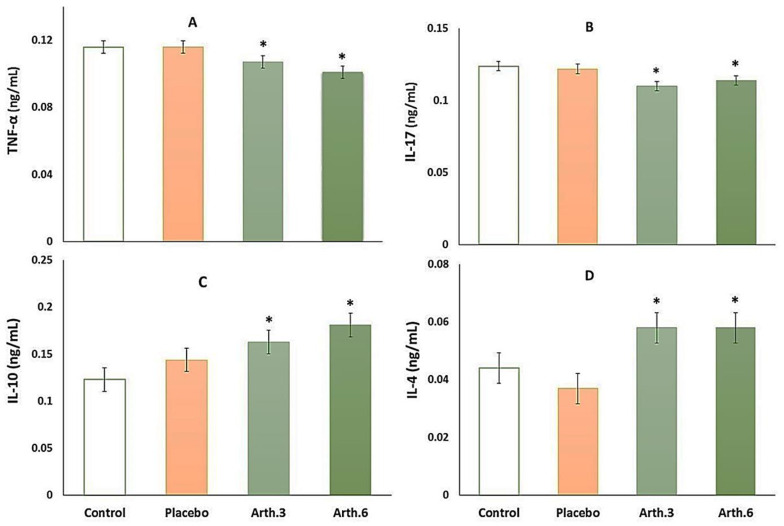
Serum TNF-α (**A**), IL-17 (**B**), IL-10 (**C**), and IL-4 (**D**) levels in different groups of patients with osteoarthritis (control and after taking placebo for 3 months and Arthrocen for 3 and 6 months) measured by ELISA. Data were presented as Mean ± S.E.M., N = 25. * Significant difference at P < 0.05 in comparison to the control and placebo groups according to per t-test.

**Table 1 medicina-58-00228-t001:** Examination items before and after treatment with placebo and Arthrocen.

Item	Group ^†^	Before Treatment (±S.E.M)	After Treatment (±S.E.M)	*p*-Value ^‡^
Joint internal tenderness	Placebo	0.45 ± 0.07	0.24 ± 0.05	*p* < 0.01
	Arthrocen	0.38 ± 0.07	0.08 ± 0.04	*p* < 0.001
	*p*-value	*p* = 0.6	*p* = 0.0139	-
Joint lateral tenderness	Placebo	0.34 ± 0.06	0.14 ± 0.04	*p* < 0.05
	Arthrocen	0.29 ± 0.06	0.03 ± 0.02	*p* < 0.001
	*p*-value	*p* = 0.64	*p* = 0.04	-
Joint patella crepitus	Placebo	1.29 ± 0.08	1 ± 0.05	*p* < 0.001
	Arthrocen	1.15 ± 0.06	0.9 ± 0.05	*p* < 0.001
	*p*-value	*p* = 0.16	*p* = 0.12	-
Joint shrug sign	Placebo	0.48 ± 0.09	0.16 ± 0.05	*p* < 0.001
	Arthrocen	0.38 ± 0.07	0.12 ± 0.04	*p* < 0.001
	*p*-value	*p* = 0.46	*p* = 0.59	-

^†^ *p*-value in this column represents the pre-test and post-test differences between placebo and Arthrocen groups. ^‡^ *p*-value in this column represents the difference in pre-test and post-test values within placebo and Arthrocen groups.

**Table 2 medicina-58-00228-t002:** Questionnaire items before and after treatment with placebo and Arthrocen.

Item	Group ^†^	Before Treatment (±S.E.M)	After Treatment (±S.E.M)	*p*-Value ^‡^
SF-20	Placebo	46.09 ± 15.8	48.69 ± 14.8	*p* = 0.116
	Arthrocen	45.41 ± 15.9	52.29 ± 14.9	*p* < 0.001
	*p*-value	*p* = 0.147	*p* = 0.295	-
WOMAC	Placebo	63.98 ± 16- 1	55.85 ± 18.9	*p* < 0.79
	Arthrocen	67.56 ± 15.3	56.59 ± 17.8	*p* < 0.001
	*p*-value	*p* = 0.312	*p* = 0.83	-
LISOK	Placebo	21.06 ± 0.6	20.4 ± 0.6	*p* < 0.29
	Arthrocen	21.1 ± 0.6	20.9 ± 0.6	*p* = 0.69
	*p*-value	*p* = 0.9	*p* = 0.5	-
VAS (pain)	Placebo	4.6 ± 0.3	3.76 ± 0.3	*p* < 0.01
	Arthrocen	4.4 ± 0.3	3.56 ± 0.2	*p* < 0.001
	*p*-value	*p* = 0.67	*p* = 0.66	-

^†^ *p*-value in this column represents the pre-test and post-test differences between placebo and Arthrocen groups. ^‡^ *p*-value in this column represents the difference in pre-test and post-test values within placebo and Arthrocen groups.

## Data Availability

Not applicable.

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
