# Peer review of "Joint Dysfunctionality Alleviation along with Systemic Inflammation Reduction Following Arthrocen Treatment in Patients with Knee Osteoarthritis: A Randomized Double-Blind Placebo-Controlled Clinical Trial"

_medicina, 2022, doi:10.3390/medicina58020228_

Round 1

Reviewer 1 Report

BRIEF SUMMARY

This was an RCT, where authors investigated the efficacy as well as the effects of avocado/soy unsaponifiables (ASU or Arthrocen), on cytokines changes in patients with knee OA. Authors report that Arthrocen treatment was associated with a significant reduction in global WOMAC and a significant increment of SF20 within but not between groups.

I congratulate the authors on their work. This is a good paper with informative figures and tables. The topic is timely and clinically important. The paper contributes to the clinical decision-making for people with knee OA. However before it can be published, I suggest authors consider my points below.

SPECIFIC COMMENTS

TITLE

Clinical efficacy and biomedical changes”: this is somewhat ambiguous, please be more specific.

ABSTRACT

Both patients and physicians pain assessments were significantly improved””. Please be more specific about what do you mean by improved i.e. increased, decreased, etc, here and throughout the paper.

Please rephrase conclusions-this is just a repetition of the results.

INTRODUCTION

Line 37: “demolition”: is there a better word instead of this?

Line 71: “While inflammation plays a major role in the pathology of OA….”. This need references and I suggest providing the following studies to demonstrate the association between inflammation and presence of knee OA ( https://pubmed.ncbi.nlm.nih.gov/23508866/ ) and effects on pain, physical function (https://ard.bmj.com/content/63/2/200 ) and muscle weakness (https://pubmed.ncbi.nlm.nih.gov/28929165/ )

In the current form, it is quite difficult to figure out from the information flow in the introduction, why it is important to study this, what is the added value of this paper to current knowledge, and who will benefit from this. Please clarify.

METHODS

The whole section (and actually the paper) should be structured according to relevant reporting guidelines for RCTs such as CONSORT. Information on study design, setting, the definition of outcomes, instrumentation, data postprocessing, etc is currently missing or superficially touched upon and should be provided in separate paragraphs to facilitate reading. In the current form, it is difficult to judge what exactly was done. This is a major concern.

https://www.equator-network.org/reporting-guidelines/consort/

RESULTS

A flow diagram of the study procedure should be presented.

The presentation of the results should be provided in subsections depending on the outcomes or type of data analysis i.e within/between groups.

Please provide a description of your study population stratified by the treatment group (age, sex, BMI etc)

DISCUSSION

Line 197-202: this is an unnecessary repetition of information provided in the intro. I suggest starting off the discussion with reminding the reader about the study objectives and summarizing the findings.

The discussion lacks a paragraph discussing potential mechanisms underlying the beneficial effects of the treatment of the outcomes.

The discussion lacks a paragraph discussing potential clinical implications of your findings. Particularly, about using the ASU as an adjunct method to other established interventions in people with knee OA such as lifestyle-modification (https://pubmed.ncbi.nlm.nih.gov/11567539/ ), exercise (https://www.ncbi.nlm.nih.gov/pmc/articles/PMC3635671/ ), physical modalities ( https://pubmed.ncbi.nlm.nih.gov/25162407/ ) knee orthoses (https://pubmed.ncbi.nlm.nih.gov/30099859/ ) etc.

The discussion lacks a paragraph regarding the limitations of the study.

CONCLUSION

This is just a repetition of the results. Please rephrase and summarize the clinical/practical value of your findings.

Author Response

Reviewer 1

This was an RCT, where authors investigated the efficacy as well as the effects of avocado/soy unsaponifiables (ASU or Arthrocen), on cytokines changes in patients with knee OA. Authors report that Arthrocen treatment was associated with a significant reduction in global WOMAC and a significant increment of SF20 within but not between groups.

I congratulate the authors on their work. This is a good paper with informative figures and tables. The topic is timely and clinically important. The paper contributes to the clinical decision-making for people with knee OA. However before it can be published, I suggest authors consider my points below.

SPECIFIC COMMENTS

TITLE

-“Clinical efficacy and biomedical changes”: this is somewhat ambiguous, please be more specific.

Response: Thank you. The title was modified as “Joint pain and dysfunctionality alleviation along with systemic inflammation reduction following Arthrocen treatment in patients with knee osteoarthritis: A Randomized Double-Blind Placebo-Controlled Clinical Trial” in the text.

ABSTRACT

-“Both patients and physicians pain assessments were significantly improved””. Please be more specific about what do you mean by improved i.e. increased, decreased, etc, here and throughout the paper.

Response: The comment was considered in the Abstract section in highlighted lines as follows: Both pain sensation and scored by specialists (VAS), respectively, quality of life indices including WOMAC and SF-20 as well as joint dysfunctionality symptoms assessed by physicians, were significantly improved (P< 0.05) after Arthrocen treatment in OA patients. The serum levels of anti-inflammatory interleukins 4 and 10 were also augmented while levels of inflammatory IL-17 and TNF-É‘ cytokines were decreased significantly (P<0.05) compared with the control groups during the 3 and 6-month treatment.

- Please rephrase conclusions-this is just a repetition of the results.

Response: The rephrase conclusions was done in Abstract section as follows: Arthrocen consumption may increase the quality of life in OA patients through amelioration of inflammation and improvement of functional activities without any long-term adverse effects.

INTRODUCTION

-Line 37: “demolition”: is there a better word instead of this?

Response: It was changed to destruction in the text.

-Line 71: “While inflammation plays a major role in the pathology of OA….”. This need references and I suggest providing the following studies to demonstrate the association between inflammation and presence of knee OA (https://pubmed.ncbi.nlm.nih.gov/23508866/ ) and effects on pain, physical function (https://ard.bmj.com/content/63/2/200 ) and muscle weakness (https://pubmed.ncbi.nlm.nih.gov/28929165/ )

In the current form, it is quite difficult to figure out from the information flow in the introduction, why it is important to study this, what is the added value of this paper to current knowledge, and who will benefit from this. Please clarify.

Response: Thank you. the statement about current comment was added into the Introduction in highlighted lines as follows: In terms of cellular mechanisms of OA, it is mainly started by sequential inflammation, and then local inflammation in the damaged joints that is associated with systemic inflammation markers [1]. It is proposed that systemic inflammation has a key role in osteoarthritis progression that is associated with pain severity, muscle weakness, and poor proprioception in patients’ OA  [2,3].

  1. Chow YY, Chin K-Y. The role of inflammation in the pathogenesis of osteoarthritis. Mediators of inflammation 2020, 2020, 8293921.
  2. Cudejko, T.; Van der Esch, M.; Van der Leeden, M.; Holla, J.; Roorda, L.D.; Lems, W.; Dekker, J. Proprioception mediates the association between systemic inflammation and muscle weakness in patients with knee osteoarthritis: results from the Amsterdam Osteoarthritis cohort. Journal of rehabilitation medicine 2018, 50, 67-72.
  3. Stürmer, T.; Brenner, H.; Koenig, W.; Günther, K. Severity and extent of osteoarthritis and low grade systemic inflammation as assessed by high sensitivity C reactive protein. Annals of the rheumatic diseases 2004, 63, 200-205.

METHODS

- The whole section (and actually the paper) should be structured according to relevant reporting guidelines for RCTs such as CONSORT. Information on study design, setting, the definition of outcomes, instrumentation, data postprocessing, etc is currently missing or superficially touched upon and should be provided in separate paragraphs to facilitate reading. In the current form, it is difficult to judge what exactly was done. This is a major concern.

https://www.equator-network.org/reporting-guidelines/consort/

Response: Thank you very much the whole of methodology was rearranged and separated according CONSORT platform and addressed in the text in the yellow lines.

RESULTS

- A flow diagram of the study procedure should be presented.

Response: The comment was done as Figure 1.

- The presentation of the results should be provided in subsections depending on the outcomes or type of data analysis i.e within/between groups.

Response: The method and result sections were provided in subsections according to type of analysis and methods.

- Please provide a description of your study population stratified by the treatment group (age, sex, BMI etc)

Response: We have decided to present some of the above data elsewhere at the first, which has already been had in the questionnaires. We therefore request that these data not be included in this study, if it is possible.

DISCUSSION

- Line 197-202: this is an unnecessary repetition of information provided in the intro. I suggest starting off the discussion with reminding the reader about the study objectives and summarizing the findings.

Response: Thank you very much. the comment was considered at the start of the discussion about the study objectives and summarizing the findings in the highlighted lines.

- The discussion lacks a paragraph discussing potential mechanisms underlying the beneficial effects of the treatment of the outcomes.

Response: The comment was considered in the discussion and conclusion sections as highlighted lines or modification of the revision.

- The discussion lacks a paragraph discussing potential clinical implications of your findings. Particularly, about using the ASU as an adjunct method to other established interventions in people with knee OA such as lifestyle-modification (https://pubmed.ncbi.nlm.nih.gov/11567539/), exercise (https://www.ncbi.nlm.nih.gov/pmc/articles/PMC3635671/ ), physical modalities ( https://pubmed.ncbi.nlm.nih.gov/25162407/ ) knee orthoses (https://pubmed.ncbi.nlm.nih.gov/30099859/ ) etc.

Response: I should respond that in this study, we did not involve in current subjects that can be interested for future studies using ASU intervention on these concepts. However, the comment’s issues were reflected in the conclusion section for better justification about ASU benefits as follows: These findings reasonably advocate the anti-inflammatory properties of Arthrocen that may affect the joint pain and dysfunctionality improvement as well as the quality of life in OA patients. Improving joint function and pain relief can cause patient activity, which can be important public health concerning the recovery of quality in lifestyle. At the clinical level, Arthrocen may prove to be an effective therapeutic agent with minimum side effects that prevents progression of OA symptoms.

- The discussion lacks a paragraph regarding the limitations of the study.

Response: Thank you very much. The limitations were added to the discussion section in highlighted lines as follows:

The limitations of our study would be taken into account when interpreting our results. At first, the interference events like smoking and pollution may involve the assessment of systemic inflammation or pain when using the measurements in long term within a person. Secondly, although widely used to assess the intensity of pain in clinical studies, the measurement of pain using a VAS is influenced by a variety of factors associated with the perception of pain [1,2]. Nevertheless, pain is always subjective and the observed associations provide evidence for this subjective phenomenon to be better associated with the markers of systemic inflammation than any of the objective markers of OA assessed in our study. Thirdly, the lack of local inflammatory characteristics is a limitation in which neither knee joint nor synovitis biomarkers were assessed. Finally, the pathology behind inflammation in additional joint regions remains indefinable.

  1. Cudejko, T.; Van der Esch, M.; Van der Leeden, M.; Holla, J.; Roorda, L.D.; Lems, W.; Dekker, J. Proprioception mediates the association between systemic inflammation and muscle weakness in patients with knee osteoarthritis: results from the Amsterdam Osteoarthritis cohort. Journal of rehabilitation medicine 2018, 50, 67-72.
  2. Stürmer, T.; Brenner, H.; Koenig, W.; Günther, K. Severity and extent of osteoarthritis and low grade systemic inflammation as assessed by high sensitivity C reactive protein. Annals of the rheumatic diseases 2004, 63, 200-205.

CONCLUSION

This is just a repetition of the results. Please rephrase and summarize the clinical/practical value of your findings.

Response: The conclusion was modified following statements of “These findings reasonably advocate the anti-inflammatory properties of Arthrocen that may affect the joint pain and dysfunctionality improvement as well as the quality of life in OA patients. Improving joint function and pain relief can cause patient activity, which is important public health concerning the recovery of quality and length of life. At the clinical level, Arthrocen may prove to be an effective therapeutic agent with minimum side effects that prevent progression of OA symptoms.” in the text.

Reviewer 2 Report

The present work is one of  many presented to explain the effects of neutracetic materials now uptodate in the scientific literature in order to better qualify  their use .In this optics the present work is in line with the parameters used in the scientific literature and is well done,the groups of patients is well defined and appropriate ,the scientific design well proposed and the parameters involved well appropriate .In particular measurements of proinflammatory and inflammatory cytokines in patients treated with 300 mg/day ASU for tree months give a  plausible explanation on the small effects on the long term administration of this neutracetic material on OA.

Author Response

Reviewer 2

Comments and Suggestions for Authors

- The present work is one of  many presented to explain the effects of neutracetic materials now uptodate in the scientific literature in order to better qualify  their use .In this optics the present work is in line with the parameters used in the scientific literature and is well done,the groups of patients is well defined and appropriate ,the scientific design well proposed and the parameters involved well appropriate .In particular measurements of proinflammatory and inflammatory cytokines in patients treated with 300 mg/day ASU for tree months give a  plausible explanation on the small effects on the long term administration of this neutracetic material on OA.

Response: Thank you very much for your comments. In line with the respected referee, ASU shows a middle or small anti-inflammatory effects in compared with standard drugs like corticosteroids. However, it is important note that reports about ASU not only have not shown still any side effects in long term administration but also it could heal the chondrocyte tissue damage resulting in joint repair and control of inflammation progression [1].

  1. Christiansen BA, Bhatti S, Goudarzi R, Emami S (2015) Management of Osteoarthritis with Avocado/Soybean Unsaponifiables. Cartilage 6:30-44. doi:10.1177/1947603514554992

Reviewer 3 Report

The topic is interesting as the intervention is safe and requires little effort.

Abstract:

In the abstract, it is stated that three Visual Analogue Scales were used as outcomes – please state what was measured on these scales.

In the abstract, please provide the mean pain results, not just the significance levels.

Introduction section:

Please provide additional information regarding the working mechanisms of the intervention. If I remember correctly, avocados contain some of the same fatty acids in olive oil and these substances have shown to reduce inflammation through an activation of the sirtuin genes.

Method section:

Was the study reported in adherence to a reporting checklist, such as CONSORT? Please indicate whether the methods were registered a priori on a website, such as on the webpage www.clinicaltrials.gov.

Discussion section:

I strongly recommend adding a limitations section to the discussion section. If an a priori protocol was not provided, please state this in the limitations section.

Author Response

Reviewer 3

Comments and Suggestions for Authors

The topic is interesting as the intervention is safe and requires little effort.

 Abstract:

- In the abstract, it is stated that three Visual Analogue Scales were used as outcomes – please state what was measured on these scales.

Response: It was reflected by …three Visual Analog Scales (VASs) as pain quality indices, in the abstract section. Also, more details about VASs have been had in the method and results sections.

- In the abstract, please provide the mean pain results, not just the significance levels.

 Response: The comment was considered in the abstract in yllow lines as follows: Both pain sensation and scored by specialists (as VASs), respectively, quality of life indices including WOMAC and SF-20 as well as joint dysfunctionality symptoms assessed by physicians, were significantly improved (P< 0.05) after Arthrocen treatment in OA patients.

Introduction section:

- Please provide additional information regarding the working mechanisms of the intervention. If I remember correctly, avocados contain some of the same fatty acids in olive oil and these substances have shown to reduce inflammation through an activation of the sirtuin genes.

Response: A complete description about ASU structure and mechanisms related to anti-inflammatory and healing effects on OA has been reflected in the introduction and discussion sections. Also, in agree with the referee, I should mention that phytosterols and fatty acids are the major components of avocado and soy that can find in other vegetable sources like olive oil that have anti-inflammatory effects. However, our knowledge is not in levels of molecular genomics of the substances.

 Method section:

- Was the study reported in adherence to a reporting checklist, such as CONSORT? Please indicate whether the methods were registered a priori on a website, such as on the webpage www.clinicaltrials.gov.

 Response: Also, as kind reminder, within our current paper, we stated that the study protocol was evaluated and approved by the Ethical Committee of the Ministry of Health & Education (Tehran, Iran). Therefore, it makes sense for using IRAN Ministry of Health Consort checklist for the current study.  For better clarification, the following statement was added to the method section in highlighted line:  In our research we used Consolidated Standards of Reporting Trials (CONSORT) that was adapted from www.CONSORT-statement.org.

Discussion section:

- I strongly recommend adding a limitations section to the discussion section. If an a priori protocol was not provided, please state this in the limitations section.

Response: Thank you very much. The limitations were added to the discussion section in highlighted lines as follows:

The limitations of our study would be taken into account when interpreting our results. At first, the interference events like smoking and pollution may involve the assessment of systemic inflammation or pain when using the measurements in long term within a person. Secondly, although widely used to assess the intensity of pain in clinical studies, the measurement of pain using a VAS is influenced by a variety of factors associated with the perception of pain [1,2]. Nevertheless, pain is always subjective and the observed associations provide evidence for this subjective phenomenon to be better associated with the markers of systemic inflammation than any of the objective markers of OA assessed in our study. Thirdly, the lack of local inflammatory characteristics is a limitation in which neither knee joint nor synovitis biomarkers were assessed. Finally, the pathology behind inflammation in additional joint regions remains indefinable.

  1. Cudejko, T.; Van der Esch, M.; Van der Leeden, M.; Holla, J.; Roorda, L.D.; Lems, W.; Dekker, J. Proprioception mediates the association between systemic inflammation and muscle weakness in patients with knee osteoarthritis: results from the Amsterdam Osteoarthritis cohort. Journal of rehabilitation medicine 2018, 50, 67-72.
  2. Stürmer, T.; Brenner, H.; Koenig, W.; Günther, K. Severity and extent of osteoarthritis and low grade systemic inflammation as assessed by high sensitivity C reactive protein. Annals of the rheumatic diseases 2004, 63, 200-205.

Round 2

Reviewer 1 Report

Authors fully addressed my comments.

Reviewer 3 Report

The manuscript has been revised adequately.